# Crack-Tip Opening Displacement of Girth Welds in a Lean X70 Pipeline Steel

**DOI:** 10.3390/ma17020391

**Published:** 2024-01-12

**Authors:** Jing Li, Peng Yu, Nitin Saini, Leijun Li

**Affiliations:** 1Department of Chemical and Materials Engineering, University of Alberta, Edmonton, AB T6G 2V4, Canada; 2Engineering Technology Training Center, Nanjing Vocational University of Industry Technology, Nanjing 210000, China

**Keywords:** X70, fracture toughness, girth weld, ferrite morphology

## Abstract

Crack-tip opening displacement (CTOD) tests were conducted on girth welds of two API 5L X70 pipeline steels (pipe A and pipe B) to investigate the influence of base metal composition on the fracture toughness of the joint. CTOD measurements across the weld showed that the weld fusion zone had the lowest CTOD values for both pipes, with pipe B having a higher CTOD value than pipe A. Detailed microstructure characterization of the multi-pass weld showed that the fusion zone in both pipes consisted of three distinct zones: the columnar zone, the coarse equiaxed zone, and the fine equiaxed zone. Both the columnar zone and coarse-grained equiaxed zone had acicular ferrite and grain boundary ferrite microstructures, whereas the fine-grained equiaxed zone had a finer ferrite microstructure compared to the other two zones. The main difference between the two pipes was the variation in ferrite grain sizes and the volume fractions of grain boundary ferrite and acicular ferrite. In comparison to pipe B, pipe A, with a higher concentration of Mo, Ni, and Cu in both the base metal and the weld fusion zones, consisted of a higher volume fraction of grain boundary ferrite and a lower volume fraction of acicular ferrite in the columnar and coarse-grained equiaxed zones. The lower concentration of Mo, Ni, and Cu in pipe B likely resulted in the formation of a predominantly acicular ferrite microstructure in the fusion zone, thereby improving the toughness of the weld joint in comparison to pipe A.

## 1. Introduction

Over the past several decades, steel pipelines have provided one of the most economical transportation means for crude oil and gas. The long-distance pipelines are exposed to severe service conditions such as earthquakes, landslides, and debris flow. API 5L X70 Grade pipeline steel (70,000 psi minimum yield strength and 82,000 psi minimum ultimate tensile strength) has become popular in the fabrication of oil and natural gas pipelines because of its low cost and excellent combination of high strength and toughness. However, this combination of strength and toughness is always deteriorated by the improper heat input and the multiple thermal cycles during multi-pass welding. Since girth welding is unavoidable in pipeline construction, the quality of the girth weld plays an essential role in ensuring the quality and safety of pipelines. For design and safety concerns, there is a pressing need to improve the fracture toughness of the weld and the heat-affected zone, both of which have been shown to have a higher sensitivity to cracking.

The fracture toughness of the weld metal is strongly affected by the composition and microstructure developed during the solidification and cooling of the weld pool. Alloying elements in the weld play an essential role in microstructural control [1]. For instance, nickel (Ni) can either decrease or increase the impact toughness, depending on its concentration in the base metal. As reported by Zhang [2], Ni increases the toughness by assisting the formation of acicular ferrite at the expense of proeutectoid ferrites such as grain boundary ferrite and ferrite side plates [3]. On the other hand, Crockett et al. [4] have shown that the addition of Ni in the range 0.02–0.87 wt.% in combination with Mo can decrease the fraction of acicular ferrite in the weld metal, thereby deteriorating the impact toughness. Bhole et al. [3] also reported that it becomes easier to form side-plate ferrite with the second phases at the expense of acicular ferrite when the weld contains Ni in combination with high levels of Mn (from 1.55 to 1.59 wt.%). The influence of copper (Cu) on microstructure and mechanical properties, reported in [5,6,7], is that by increasing copper from 0.02 to 1.4 wt.%, toughness decreased in the multi-pass shielded metal arc C-Mn weld [6,7]. Thus, the combined effect of various alloying elements in the pipeline steels can either improve or deteriorate the toughness of the weldment [3,8,9].

The purpose of this work is to understand the effect of base metal composition on the toughness of the multi-pass weld. For this purpose, crack-tip opening displacement (CTOD) tests were conducted on girth welds of two API 5L X70 pipeline steels (pipe A and pipe B) to investigate the influence of base metal composition on the fracture toughness of the joint. The analysis mainly focuses on the effect of base metal composition on the developed microstructure of the multi-pass weldment and correlating the observed microstructure to the toughness values.

## 2. Materials and Methods

Two girth-welded X70 steel pipes (pipe A and pipe B) were investigated in this research. Different suppliers were used for evaluating the effect of subtle chemical differences between suppliers. These two pipes shared identical wall thickness and outside diameter, which were 9.8 mm (0.386 inches) and 762 mm (30 inches), respectively. Both pipes were welded using the same filler metal. The chemical compositions of base metals and filler metals are given in Table 1. No post-weld heat treatment was applied for both pipe welds. One noticeable difference between pipe base metals A and B is the content of P, Ni, and Cu. Base metal A contains a significantly higher amount (around 10 times) of P, Ni, Mo, and Cu than base metal B, which makes base metal B a very lean chemistry.

Girth welds with double V-type grooves were industrially manufactured in the vertical (ASME 5 G) position with multi-pass deposits using an automatic gas–metal arc-welding (GMAW) machine using the parameters as shown in Table 2. Crack-tip opening displacement (CTOD) tests were conducted at −5 °C in accordance with the methods outlined in CSA Z662-15 Annex K [10] and ISO 15653:2018 [11] to obtain the fracture toughness data. CTOD tests were performed on specimens machined from the pipes with B × 2B geometry, as shown in Figure 1. Square bar specimens were notched on the weld centerline or heat-affected zone (HAZ), as specified by CSA Z662-15. To investigate the relationship between microstructure and fracture toughness, the broken CTOD specimens were cut and polished using standard metallographic practices. Firstly, the metallographic specimens were cut at the root of the pre-crack along the XY plane and polished to a 0.05 μm finish using colloidal silica. Then the polished specimens were etched in a freshly prepared 2% Nital solution for microstructure examinations along the crack initiation plane (XY plane, refer to Figure 1). Finally, the studied specimens were cut in the middle of the X direction along the YZ plane (refer to Figure 1) to characterize the microstructure near the crack path. Microstructure was characterized using an optical Olympus BX61 microscope and a Zeiss Sigma field emission-scanning electron microscope (FE-SEM).

Microhardness measurements across the weld were performed using a load of 0.5 kgf, a dwell time of 10 s, and an interspacing of 150 μm between indentations. The metallographic measurement method was used to calculate the dilution level for the weld fusion zone by the base metal. The grain size and the diameter of fracture dimples were measured using the linear intercept method under the ImageJ software v1.54f.

## 3. Results

### 3.1. Microstructure of the As-Welded Pipe

Macrographs and the microstructures of the weld fusion zone along the YZ plane (refer to Figure 1) of both pipes are shown in Figure 2. It can be seen from the macrograph in Figure 2a,b that except for the last weld pass, each weld pass was reheated by the subsequent pass, resulting in three different kinds of tempered microstructure in the weld fusion zone. Firstly, the columnar zone consisted of columnar grains that pointed toward the weld centerline. The microstructure of the columnar zone in both pipes (Figure 2c,d) consisted of grain boundary ferrite and acicular ferrite. In comparison with pipe B, the fusion zone of pipe A had a larger fraction of grain boundary ferrite along the prior columnar austenite grain boundaries (refer to Figure 2c,d). Secondly, the equiaxed grain zone, which consisted of equiaxed grains with a microstructure similar to the columnar zone, consisted of grain boundary ferrite and acicular ferrite (Figure 2c,d). Thirdly, the fine-grained equiaxed zone of both pipes had a denser and finer distribution of acicular ferrite, as shown in Figure 2g,h. It should also be noted that the heat-affected zone (HAZ) of pipe A was much wider (2.36 mm) in comparison to pipe B (1.73 mm) (refer to Figure 2a,b).

The average proportions of the three distinct microstructure zones along the weld centerlines of pipe A and pipe B were measured. As presented in Table 3, pipe A, with a lower CTOD value, had a higher proportion of columnar zone (70.6%) compared with pipe B (65.2%) and lower proportions of the coarse-grained equiaxed zone (20.1%) in comparison with pipe B (25.4%). In addition, proportions of fine-grained equiaxed zone were found along the weld centerlines of both pipes, which were 9.3% and 9.4%, respectively.

### 3.2. CTOD and Hardness Distribution across the Weld

The CTOD results for both pipes are shown in Figure 3a. For both pipes, the specimen with a notch located in the weld centerline showed the lowest CTOD value. The lowest CTOD value obtained in the weld of pipe A was 0.31 mm, which was about 28% lower than that of pipe B (0.43 mm). Vickers microhardness profiles across the weld measured at the middle of specimens, as indicated in Figure 3c, are shown in Figure 3b. A similar general trend of variation in hardness was observed for both pipes. The coarse-grained heat-affected zone (CGHAZ) exhibited the highest hardness, while the fine-grained heat-affected zone (FGHAZ) exhibited the lowest hardness. The weld fusion zone and the base metal had an intermediate hardness. The average hardness of base metal for pipe B (215 HV) was about 6.5% lower than that of pipe A (230 HV).

### 3.3. Fracture Behavior

#### 3.3.1. Microstructure along the Crack Path

After the CTOD tests, the crack path microstructure and the fracture surfaces of both pipes were studied. Figure 4 shows the representative crack propagation microstructure, which was obtained from pipe A and B specimens with the lowest CTOD value. The crack of this specimen extended toward the grain boundary ferrite and propagated along them, as marked by the arrows in Figure 4. It appears that the grain boundary ferrite acted as the preferential crack propagation path.

#### 3.3.2. Fractography

Figure 5 shows the fracture surface morphology of the fractured CTOD specimens. Both pipes exhibited a typical quasi-cleavage pattern characterized by cleavage facets mixed with spherical dimple regions. In comparison to pipe A, pipe B, with a higher CTOD value, showed denser dimple regions with smaller cleavage facets. Pipe A had shallow dimples (refer to Figure 5c) with an average diameter of 7.7 ± 0.5 μm, while pipe B had deeper dimples with an average diameter of 4.4 ± 0.4 μm. In addition, the area fraction of dimple zones for pipe A was 91.29%, which was smaller than that of pipe B with a dimple zone fraction of 95.7%. The variations in the dimple size and morphology were consistent with the measured differences in the CTOD values.

## 4. Discussion

### 4.1. Effect of the Base Metal Chemical Composition on the Microstructure of Weld Metal

Since the weld metal was a mixture of the parent metal and filler metal, the chemical composition of the weld was not only determined by the individual chemical compositions of the filler metal but also by the degree of dilution from the base metal during the fusion welding. Since the chemical composition of the weld determined both the mechanical properties and the microstructure, an accurate dilution level was essential to estimating the weld fusion chemical composition.

The dilution level could be determined by two methods: chemical analysis and metallographic measurements. For the chemical analysis, the chemical composition of weld metal could be obtained via an electron probe micro-analyzer (EPMA). By measuring the elemental compositions of the base metal, filler metal, and weld metal, the degree of mixing (namely, the dilution level) could be calculated. Good agreement was reported to be seen between the two kinds of measurement, as mentioned in detail in reference [12]. As to metallographic dilution measurements, dilution could be understood with the schematic diagram in Figure 6 for the cross-sectional area of the girth weld of pipe B. The melted base metal *A_bm_* was the area between the fusion lines and the black dash lines, as marked by the black arrows, and the deposited filler metal *A_fm_* was the area that is surrounded by fusion lines. The dilution level D could be calculated using the following equation:(1)$$D=\frac{A_{bm}}{A_{bm}+A_{fm}}$$

The concentration of alloying element *i* in the fully mixed weld metal $C_{fz}^{i}$ was given by the following:(2)$$C_{fz}^{i}={DC}_{bm}^{i}+{\left(1-D\right)C}_{fm}^{i}$$
where $C_{bm}^{i}$ and $C_{fm}^{i}$ were the concentrations of alloying element *i* in the base metal and filler metal, respectively.

The dilution levels of some of the alloying elements in the weld metal of both pipes are shown in Table 4. The as-diluted chemical compositions of the welds containing a mixture of base metal and filler metal are shown in Table 1. For pipe A, the Ni, Mo, and Cu contents in the base metal were higher than for pipe B. Therefore, the weld of pipe A had a higher concentration of Ni, Mo, and Cu (refer to Table 4). It was reported that Ni was deleterious for cleavage fracture resistance at high Mn concentrations (more than 1.4 wt.%) and beneficial at low Mn concentrations [9], and the addition of Ni would decrease the amount of acicular ferrite. Also, Ni in combination with Cu and Mo would increase the hardness of the weld, thereby making it more susceptible to cracking and eventually reducing the toughness [4,7] of the weld.

### 4.2. Effect of the Heat Input and Microstructure on the CTOD Properties

The weld thermal cycles played an essential role in the multi-pass welding microstructure. As evident from Figure 2a,b, the HAZ of pipe A was 2.36 mm, which was 36.4% wider than that of pipe B (1.73 mm). This indicated that a higher heat input was applied during the welding process for the girth weld of pipe A. Considering the wide allowable range for welding current (refer to Table 2), it appeared that a higher welding current might have been used for pipe A. The higher heat input in pipe A in comparison to pipe B caused grain coarsening in the weld of pipe A. It was shown that the toughness was significantly influenced by the microstructural constituents [13,14]. As illustrated in Figure 2c–f, for the columnar zone and coarse-grained equiaxed zone, pipe A had a lower fraction of acicular ferrite and a higher fraction of grain boundary ferrite (13.48%), and the ferrite grain size was coarser in comparison to pipe B. It should also be noted that pipe B had a higher volume fraction of acicular ferrite and a lower volume fraction of grain boundary ferrite (7.71%). The difference in the acicular ferrite fraction could be attributed to the presence of a higher amount of nickel/molybdenum in pipe A.

The fine-grained equiaxed zone of both pipes had a denser and finer distribution of acicular ferrite, as shown in Figure 2g,h. Since acicular ferrite had a higher fraction of high-angle grain boundaries and nucleates intragranularly, forming an interlocking morphology, the acicular ferrite microstructure was more capable of arresting the propagation of a crack [15,16,17,18,19]. Also, from the crack propagation cross-section microstructure in Figure 4b,d, the crack preferentially propagated along the grain boundary ferrite. During the crack propagation process, a relatively large grain size was detrimental to crack resistance and thus would have a lower toughness/CTOD value. Thus, the presence of a higher fraction of grain boundary ferrite and a lower fraction of acicular ferrite in pipe A was the likely reason for the lower toughness of pipe A in comparison to pipe B.

Proportions of the three different kinds of microstructure along the weld centerlines, shown in Table 3, revealed that the weld with a lower proportion of columnar zone showed better fracture toughness.

## 5. Conclusions

Crack-tip opening displacement (CTOD) tests were conducted on the girth welds of two API 5L X70 pipeline steels (pipe A and pipe B) to investigate the influence of base metal composition on the fracture toughness of the joint. CTOD measurements across the weld showed that the weld fusion zone had the lowest CTOD value. Based on the results presented in this study, the following conclusions could be made:The fusion zone in both pipes consisted of three zones with different microstructures. Firstly, the columnar zone consisted of grain boundary ferrite and acicular ferrite. Secondly, the coarse-grained equiaxed zone, which consisted of equiaxed grains with a microstructure like the columnar zone, consisting of grain boundary ferrite and acicular ferrite, and finally, the fine-grained equiaxed zone with a denser and finer distribution of acicular ferrite.In comparison to pipe B, pipe A, with a higher Ni, Mo, and Cu content, likely resulted in the formation of a fusion zone microstructure with a lower fraction of acicular ferrite and a higher fraction of grain-boundary ferrite.A relatively high heat input for pipe A resulted in the formation of coarser grain-boundary ferrite in comparison to pipe B.During the crack propagation process, the presence of a higher fraction of grain-boundary ferrite and a lower fraction of acicular ferrite in pipe A was the likely reason for the lower toughness of pipe A in comparison to pipe B.

These findings indicated the importance of base metal selection with a suitable chemical composition to achieve the desired toughness of the weld joint.

## Figures and Tables

**Figure 1 materials-17-00391-f001:**
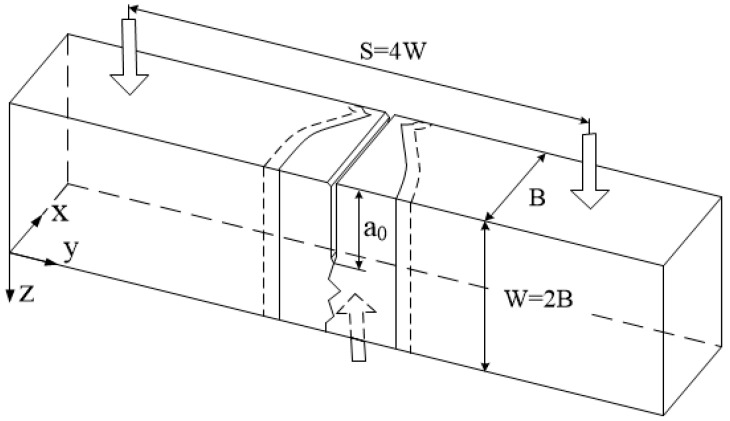
Schematic diagram of CTOD specimen with a notch located on the weld centerline. B = 8.9 mm, a_0_/W = 0.5.

**Figure 2 materials-17-00391-f002:**
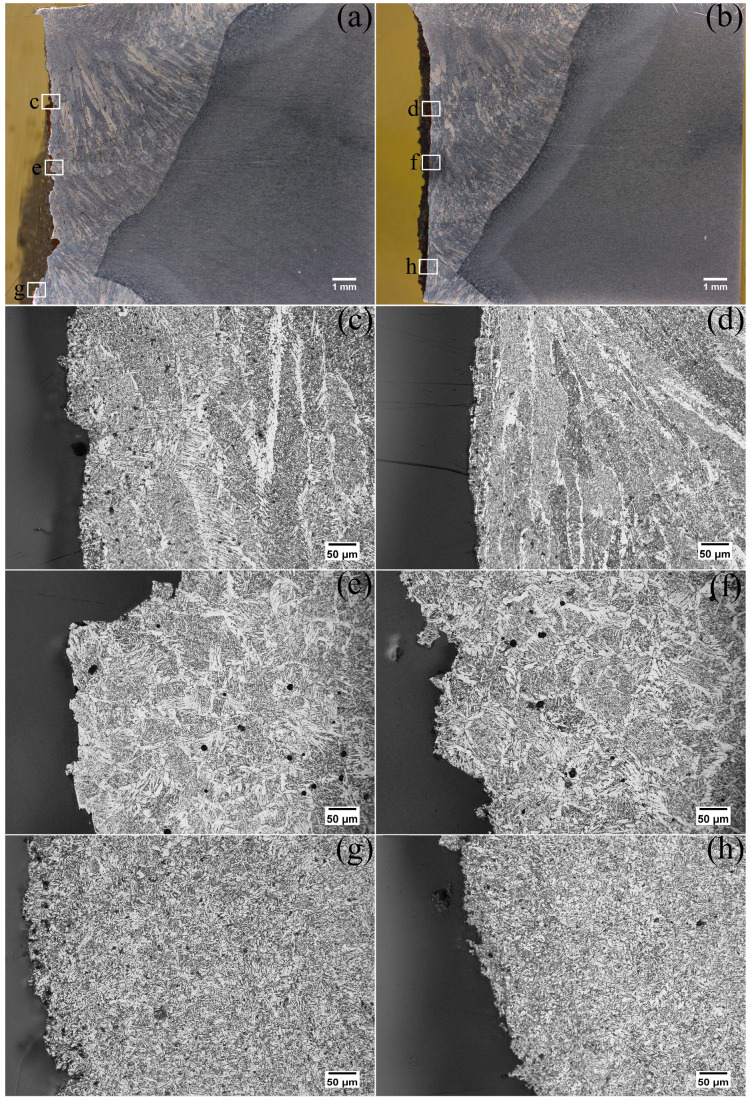
Macrographs of CTOD crack propagation looking at the z-direction, as shown in Figure 1 for (**a**) pipe A and (**b**) pipe B. The (**c**,**d**), (**e**,**f**), and (**g**,**h**) are columnar grain zones, coarse-grained equiaxed zones, and fine-grained equiaxed zones, respectively, for pipe A and pipe B.

**Figure 3 materials-17-00391-f003:**
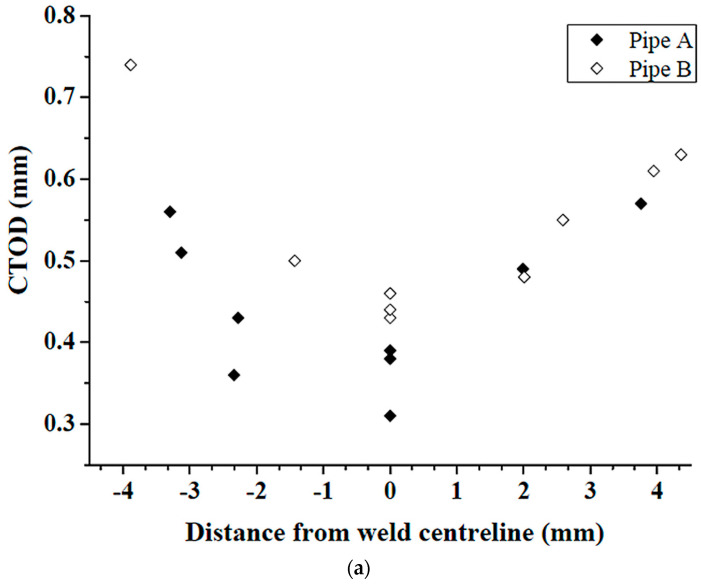
(**a**) CTOD results obtained from specimens with notches located in the weld centerline or HAZ for pipes A and B; (**b**) hardness variations at the middle height of the two girth welds; and (**c**) cross-section schematic diagram of CTOD specimens with a notch located on the weld centerline. The solid lines indicate the weld fusion boundaries, and the dashed lines indicate the boundaries between the heat-affected zone and the base metal.

**Figure 4 materials-17-00391-f004:**
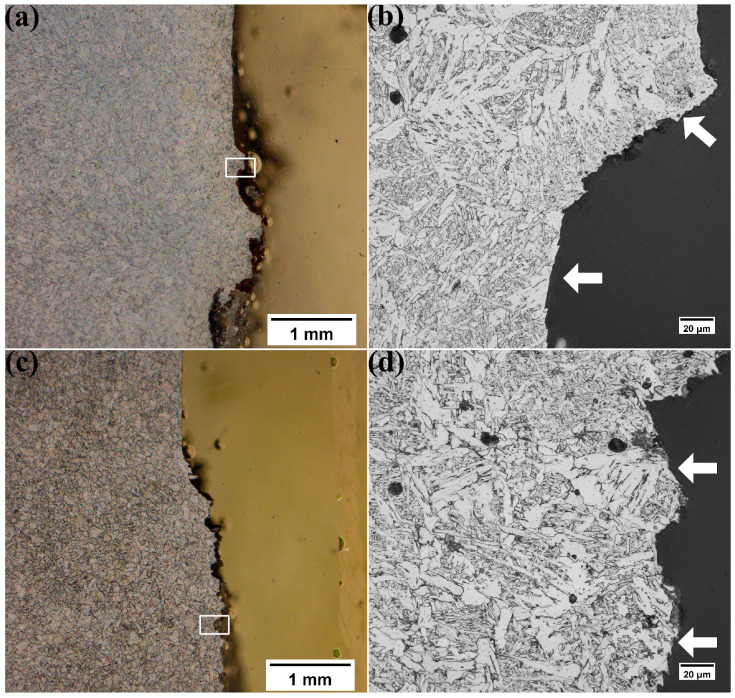
Micrographs looking at the x direction shown in Figure 1 of the CTOD crack propagation and the magnified square boxes of pipe A (**a**,**b**) and pipe B (**c**,**d**) with a notch located in the weld centerline. The white arrows indicate the locations of side-plate ferrite.

**Figure 5 materials-17-00391-f005:**
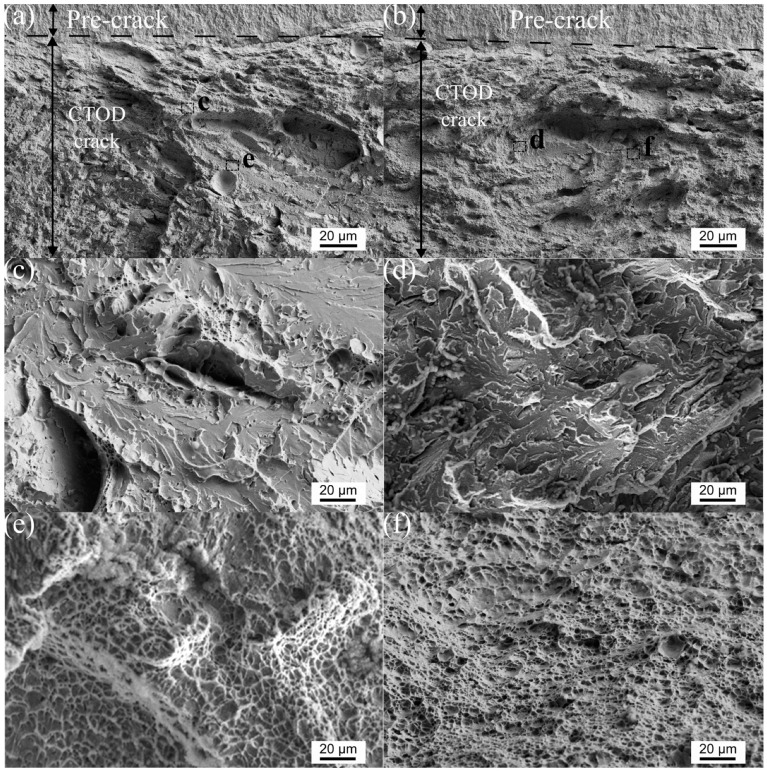
SEM examination of the fracture surfaces of the CTOD tested specimens with the lowest CTOD value of both pipes ((**a**,**c**,**e**) for pipe A and (**b**,**d**,**f**) for pipe B) showing quasi-cleavage fracture. (**c**,**d**) and (**e**,**f**) are the cleavage and ductile parts of each pipe, respectively.

**Figure 6 materials-17-00391-f006:**
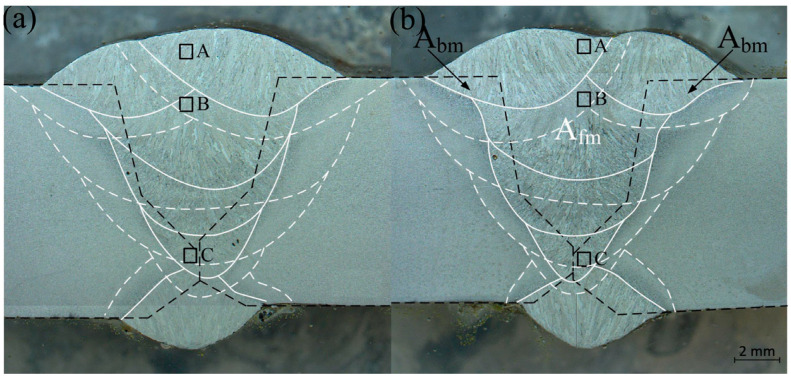
A schematic illustration of dilution exhibiting the amount of melted base metal (*A_bm_*) and deposited filler metal (*A_fm_*) in the girth weld of pipes A (**a**) and B (**b**). White solid lines are the profile of each bead, and white dashed lines are the HAZ profile of each bead. Black squares A, B, and C are columnar grain zones, coarse-grained equiaxed zones, and fine-grained equiaxed zones, respectively.

**Table 1 materials-17-00391-t001:** Chemical composition (wt.%) of the base metal, filler metal, and weld metal of the investigated X70 pipeline steel and weld joints.

Elements	C	Si	Mn	P	S	Ni	Cr	Mo	Cu	V	Nb	Al+Ti+N+B+Ca	Fe
Pipe A	0.053	0.29	1.72	0.1	0.001	0.1	0.25	0.12	0.23	0.003	0.079	0.403	Bal.
Pipe B	0.07	0.18	1.56	0.014	0.001	0.01	0.23	0	0.02	0.04	0.07	0.442	Bal.
Filler metal	0.08	0.98	1.45	0.012	0.01	0	0.05	0.05	0.04	0.001	0	0	Bal.
Weld A	0.072	0.783	1.527	0.037	0.007	0.029	0.107	0.07	0.094	0.008	0.023	0.125	Bal.
Weld B	0.077	0.757	1.481	0.013	0.007	0.003	0.1	0.036	0.034	0.018	0.019	0.132	Bal.

**Table 2 materials-17-00391-t002:** GMAW welding parameters.

Pass	Interpass Temperature (°C)	Voltage (V)	Current (A)	Travel Speed (ipm)	Heat Input (kJ/mm)
Root Pass	48	19	185	29	0.29
Other Passes	54–73	18–21	161–245	15–54	0.23–0.44

**Table 3 materials-17-00391-t003:** The average proportions of three different kinds of microstructure along the weld centerlines of the two pipes.

Specimen	Columnar Zone	Coarse-Grained Equiaxed Zone	Fine-Grained Equiaxed Zone
Weld of pipe A	70.60%	20.10%	9.30%
Weld of pipe B	65.20%	25.40%	9.40%

**Table 4 materials-17-00391-t004:** Dilution levels and some alloy elements in the weld metal of both pipes.

Specimen	Dilution Level	Ni	Mo	Cu
Weld of pipe A	0.286	0.03	0.07	0.09
Weld of pipe B	0.279	0.003	0.036	0.03

## Data Availability

Data are contained within the article.

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
