# Peer review of "Crack-Tip Opening Displacement of Girth Welds in a Lean X70 Pipeline Steel"

_materials, 2024, doi:10.3390/ma17020391_

Round 1
Reviewer 1 Report
Comments and Suggestions for Authors
The authors of the paper presented interesting experimental results on welding of pipes using the multi-path method. The authors state that it is very important to choose the right materials for the elements to be joined (pipes) which is obvious.
I think that the paper is missing:
- an explanation of why pipes with different chemical compositions were used? Why pipes made of identical material were not used?
- Did the authors consider the possibility of controlling the weld structure, for example, using the micro-jet cooling method?
In the conclusions, the authors state that the problem is the coarseness of the weld structure in certain zones of the weld, which may be due to the chemical composition. Appropriate micro-jet cooling could affect this structure.
The presentation style and format of the article are okey. The figures are appropriate and reflect the content of the article. The article is written in a clear manner. I don't feel qualified to judge about the English language.
Author Response
The authors of the paper presented interesting experimental results on welding of pipes using the multi-path method. The authors state that it is very important to choose the right materials for the elements to be joined (pipes) which is obvious.
I think that the paper is missing:
- an explanation of why pipes with different chemical compositions were used? Why pipes made of identical material were not used?
Different suppliers were used in this application for evaluating the effect of subtle chemical differences between suppliers. This is added to Experimental Procedure for an explanation.
- Did the authors consider the possibility of controlling the weld structure, for example, using the micro-jet cooling method?
Micro-jet cooling method was not considered. The welding process and procedure parameters were selected to follow the API and CSA (ASME) standards.
In the conclusions, the authors state that the problem is the coarseness of the weld structure in certain zones of the weld, which may be due to the chemical composition. Appropriate micro-jet cooling could affect this structure.
Agree. Micro-jet cooling could have affected local tempering of the as-welded microstructure.
The presentation style and format of the article are okey. The figures are appropriate and reflect the content of the article. The article is written in a clear manner. I don't feel qualified to judge about the English language.
Reviewer 2 Report
Comments and Suggestions for Authors
The manuscript "CTOD of Girth Welds in a Lean X70 Pipeline Steel" submitted for publication on Materials has been reviewed. It deals with an experimental study on toughness of two common pipeline steels by means of CTOD tests, metallography and hardness measurements.
The manuscript appears almost clear, english acceptable despite not authors are not mother tongue.
Results are clearly presented, discussion and conclusion derived from the results.
My only concern is about the novelty aspects of the manuscript. Presented as "Article", it looks more like to a technical report rather than a research article. Materials, welding techniques and experimenatl tests are consolidated. In my opinion the novelty aspects, in relation to the state of the art, should be better highlighted both in introduction, discussion and conclusions. Which problem is this article solving? How the knowledge has been improved? These questions must be given
Furthermore the following modifications must be addressed:
1) Increase readability of Fig. 3 b);
2) Welded samples should be shown, as a form of pipe and notched samples;
3) It is not specifiied if PWHT has been performed on the samples (in case give details) or not.
After the major revisions the manuscript can be reconsidered for publication on Materials.
Comments on the Quality of English Language
Acceptable
Author Response
The manuscript "CTOD of Girth Welds in a Lean X70 Pipeline Steel" submitted for publication on Materials has been reviewed. It deals with an experimental study on toughness of two common pipeline steels by means of CTOD tests, metallography and hardness measurements.
The manuscript appears almost clear, english acceptable despite not authors are not mother tongue.
Results are clearly presented, discussion and conclusion derived from the results.
My only concern is about the novelty aspects of the manuscript. Presented as "Article", it looks more like to a technical report rather than a research article. Materials, welding techniques and experimenatl tests are consolidated. In my opinion the novelty aspects, in relation to the state of the art, should be better highlighted both in introduction, discussion and conclusions. Which problem is this article solving? How the knowledge has been improved? These questions must be given
Thanks for this comment. The paper was revised to read less like a report.
Furthermore the following modifications must be addressed:
1) Increase readability of Fig. 3 b);
The paper was entirely revised to improve readability and technical presentation.
2) Welded samples should be shown, as a form of pipe and notched samples;
Due to project proprietary restrictions, the welded samples were not available for publication. Figure 1 was shown instead.
3) It is not specifiied if PWHT has been performed on the samples (in case give details) or not.
Agree. No PWHT has been added to the manuscript in the materials and methods section.
After the major revisions the manuscript can be reconsidered for publication on Materials.
Round 2
Reviewer 2 Report
Comments and Suggestions for Authors
The manuscript has been significantly improved, all the issues have been addressed.
It can be accepted in the present form.